# PUFTAP-IoT: PUF-Based Three-Factor Authentication Protocol in IoT Environment Focused on Sensing Devices

**DOI:** 10.3390/s22187075

**Published:** 2022-09-19

**Authors:** JoonYoung Lee, JiHyeon Oh, DeokKyu Kwon, MyeongHyun Kim, SungJin Yu, Nam-Su Jho, Youngho Park

**Affiliations:** 1School of Electronic and Electrical Engineering, Kyungpook National University, Daegu 41566, Korea; 2Electronics and Telecommunications Research Institute, Daejeon 34129, Korea; 3School of Electronics Engineering, Kyungpook National University, Daegu 41566, Korea

**Keywords:** IoT, WSN, PUF, biometrics, honey list, authentication, BAN logic, ROR model, scyther

## Abstract

In IoT-based environments, smart services can be provided to users under various environments, such as smart homes, smart factories, smart cities, smart transportation, and healthcare, by utilizing sensing devices. Nevertheless, a series of security problems may arise because of the nature of the wireless channel in the Wireless Sensor Network (WSN) for utilizing IoT services. Authentication and key agreements are essential elements for providing secure services in WSNs. Accordingly, two-factor and three-factor-based authentication protocol research is being actively conducted. However, IoT service users can be vulnerable to ID/password pair guessing attacks by setting easy-to-remember identities and passwords. In addition, sensors and sensing devices deployed in IoT environments are vulnerable to capture attacks. To address this issue, in this paper, we analyze the protocols of Chunka et al., Amintoosi et al., and Hajian et al. and describe their security vulnerabilities. Moreover, this paper introduces PUF and honey list techniques with three-factor authentication to design protocols resistant to ID/password pair guessing, brute-force, and capture attacks. Accordingly, we introduce PUFTAP-IoT, which can provide secure services in the IoT environment. To prove the security of PUFTAP-IoT, we perform formal analyses through Burrows Abadi Needham (BAN) logic, Real-Or-Random (ROR) model, and scyther simulation tools. In addition, we demonstrate the efficiency of the protocol compared with other authentication protocols in terms of security, computational cost, and communication cost, showing that it can provide secure services in IoT environments.

## 1. Introduction

The rapid development of wireless networks and the Internet of Things (IoT) has created opportunities to communicate with things over the Internet. Wireless sensor networks (WSN), a combination of wireless networks and IoT sensors, are garnering increasing attention worldwide as an exciting new paradigm of IoT in various fields, such as smart home, smart city, smart transportation, and smart agriculture [1,2,3]. In this IoT-based environment, data are collected through various sensors and sensing devices, and users can access them through a gateway node. Through WSN, users can use convenient services in real-time through IoT devices in an IoT-based environment. For example, with their IoT devices, users can remotely operate the lights in their house or sprinklers in their garden.

However, because this convenient service is provided through a wireless network, it is vulnerable to illegal access by malicious attackers [4,5]. This can harm the convenience of IoT, such as invasions of user privacy and eavesdropping on privacy. Malicious attackers can also be insiders or outsiders seeking to breach network security and falsify data integrity. Moreover, problems of node and link failures (i.e., cascading failures) can occur due to the limitations of resources and energy of IoT equipment [6,7]. To this end, the development of lightweight protocols that provide secure communication between nodes and that can overcome resource and energy limitations is ongoing.

Key agreement and authentication protocols are an integral part of addressing security vulnerabilities in WSN and IoT environments and are being studied continuously. Two-factor-based authentication protocols consisting of passwords and smart cards have been proposed for secure communication in IoT-based environments [8,9,10,11,12,13,14,15,16,17,18]. However, these two-factor-based authentication protocols are also vulnerable to smart card theft and guessing attacks, among other attacks. In addition, some researchers have argued that, in two-factor authentication protocols, an attacker can guess an ID/password pair as users create easy-to-remember ID/password pair for convenience [19,20,21]. Therefore, the researchers argued that attackers can guess an ID/password pair within polynomial time. Accordingly, to respond to various attacks, including ID/password pair guessing attacks, three-factor-based authentication protocols have been proposed, involving user’s biometric information [22,23,24,25,26,27,28].

Although the three-factor-based authentication protocol is more secure than the two-factor-based authentication protocol, some researchers have found that the three-factor authentication protocols proposed in WSN and IoT environments are also not secure against multiple attacks. Although three-factor-based authentication protocols can defend against ID/password pair guessing attacks, they are still vulnerable to attacks that can be performed with values obtained through device capture attacks. Additionally, the three-factor authentication protocol is still vulnerable to replay, impersonation, and session key disclosure attacks.

In this paper, we analyze the security of two-factor-based and three-factor-based authentication protocols to discover their vulnerabilities. Chunka et al.’s protocol [16] is vulnerable to known session-specific temporary information, ID/password pair guessing, and impersonation attacks. The protocol of Amintoosi et al. [18] is also vulnerable to ID/password pair guessing attacks, thus allowing impersonation attacks. The protocol of Hajian et al. [27] is vulnerable to device physical capture attacks, and through these attacks, device impersonation and session key disclosure attacks are possible.

This paper introduces the Physical Unclonable Function (PUF) [29], which can strengthen security against device capture attacks, and honeylist [30,31], which can prevent off-line guessing and brute-force attacks from solving the vulnerabilities of three-factor-based authentication protocols. With honey_list, the authentication protocol can be secure, even if two of three factors of the authentication protocol are leaked. In addition, we configure the authentication protocol with XOR and hash functions for real-time communication of sensing devices and prevention of system down.

Therefore, this study aims to solve the security vulnerabilities of the two-factor and three-factor-based WSN authentication protocols [16,18,27]. In addition, we propose PUFTAP-IoT, a secure protocol for IoT-based environments using the three factors that are safe against various attacks in the IoT environment.

We adopt two technologies for a secure protocol for sensing devices in the IoT environment. We also invoke honey_list technology to defend against online-guessing and brute-force attacks and consider PUF to be safe against takeover attacks of sensors and sensing devices. The contributions of this paper are as follows:We prove the vulnerabilities of protocols by Chunka et al. [16] and Amintoosi et al. [18], which are two-factor authentication protocols, and Hajian et al. [27], which is a three-factor authentication protocol.PUFTAP-IoT adopts PUF [29] and honey_list [30,31] technology to be safe against various attacks. In addition, to solve the resource problem of sensors and sensing devices, only XOR and hash functions excluding elliptic curve cryptography (ECC) functions are used to lighten the protocol.Informal (non-mathematical) analysis and formal analysis are performed to prove the security of the proposed PUFTAP-IoT. Formal analysis uses the widely adopted Burrows Abadi Needham (BAN) logic [32] and Real-Or-Random (ROR) model [33]. We also use the scyther simulation tool [34] to show that PUFTAP-IoT is secure in networks over public channels.We compare PUFTAP-IoT with other authentication protocols in terms of computation cost, communication cost, and security to analyze its efficiency.

The remainder of this paper is organized as follows: Section 2 reviews two-factor and three-factor-based authentication protocols in IoT and WSN environments. Section 3 outlines the proposed system model, attacker model, PUF, fuzzy extraction, and honey list. We analyze the protocols of Zou et al., Amintoosi et al., and Hajian et al. to demonstrate security vulnerabilities. Section 5 describes PUFTAP-IoT, and the safety of PUFTAP-IoT is analyzed in Section 6. We also analyze the efficiency of the protocol in Section 7. Finally, Section 8 concludes the paper.

## 2. Related Works

Lamport [35] first proposed a password-based authentication protocol in 1981. Since then, many related studies on password-based, two-factor authentication protocols have been proposed in various network environments to protect users’ privacy. In 2009, Das [8] proposed a two-factor authentication concept using a smart card with password in an IoT-based WSN environment. Das argued that the proposed scheme has a security advantage in that it uses only a hash function to reduce communication overhead and resist various attacks. However, He et al. [9] proved [8]’s authentication protocol is vulnerable to insider attacks along with impersonation attacks in 2010. In addition, He et al. presented an improved protocol as a countermeasure against these attacks. Unfortunately, it was found by Kumar and Lee [10] that He et al.’s protocol also does not guarantee mutual authentication and cannot generate a session key. Turkanović et al. proposed a new authentication and key agreement method in the WSN environment, focusing on heterogeneous IoT. The proposed scheme allows users to negotiate session keys securely with sensor nodes using the authentication protocol. However, Amin and Biswas [12] demonstrated that the protocol of Turkanović et al. is not secure against impersonation, identity guessing, and password guessing attacks. Moreover, they showed that their scheme has an inefficient authentication phase. Amin and Biswas proposed a protocol that compensated for these problems. However, Wu et al. [13] found that the protocol of Amin and Biswas are also vulnerable to sensor capture and guessing and spoofing attacks. Shuai et al. [14] suggested an authentication protocol for smart homes in 2019. In their protocol, they use Elliptic Curve Cryptography (ECC) for efficient and anonymous authentication. They demonstrate that their protocol is secure against a variety of attacks, including desynchronization and verification table stolen attacks. However, Zou et al. [15] proved Shuai et al.’s protocol is insecure against perfect forward secrecy, node capture attack, and impersonation attacks. Moreover, they proposed more secure user authentication schemes for smart homes. In 2021, Chunka et al. [16] point out the problems with the authentication protocol for WSN environment proposed by Kalra and Sood [17]. They pointed out that the protocol proposed by Kalra and Sood is vulnerable to sensor node capture attacks and cannot provide perfect forward secrecy. In 2022, Amintoosi et al. [18] proposed a two-authentication-based authentication and key agreement protocol to ensure the privacy and security of patients’ health-related data. They claim that their protocol is safe from various attacks and is a lightweight protocol using only hash and XOR functions.

According to [19,20], people tend to choose ID/password pairs that are easy to remember. As a result, ID and password pairs are chosen from a small dictionary space. This allows an attacker to guess a user’s ID and password in polynomial time [21]. Many researchers have proposed a secure three-factor authentication scheme to prevent simultaneous ID and password pair guessing attacks.

In 2016, Amin et al. [22] proposed a three-factor authentication protocol for WSN. They designed an anonymity-preserving authentication scheme for WSN and proved that their proposed protocol is secure against multiple attacks and is more efficient than other protocols. However, Jiang et al. [23] showed that the protocol of Amin et al. is insecure against replay attacks and does not provide complete forward secrecy. To solve this security flaw, Jiang et al. presented an authentication protocol based on the Rabin cryptosystem for WSN. However, Ostad-Sharif et al. [24] demonstrate that the Jiang et al. protocol also does not provide perfect forward secrecy. In 2019, Mo et al. [25] proposed a secure three-factor-based key agreement and user authentication protocol for WSN. They presented a protocol based on ECC. They demonstrated that their protocol is able to provide security against untraceability and user anonymity. However, Yu and Park [26] pointed out that the protocol of Mo et al. is not safe for impersonation, replay, and session key disclosure attacks. Unfortunately, Hajian et al. [27] proved that the protocol proposed by Ostad-Sharif et al. [24] and Yu et al. [26] is also vulnerable to some attacks. To prevent security problems, Hajian et al. proposed a lightweight authentication protocol for IoT environments. They argued that the proposed protocol can defend against multiple attacks. In 2022, Amintoosi et al. [18] pointed out the security vulnerabilities of the authentication protocol for e-health proposed by Aghili et al. [28]. They proposed a lightweight authentication protocol for smart healthcare services that solves the security vulnerabilities of Aghili et al.’s protocol.

However, we prove that some schemes [16,18,27] are vulnerable to security attacks. We found that Chunka et al. [16] protocol is vulnerable to known session-specific temporary information, ID/password pair guessing, and impersonation attacks. Additionally, we prove that Amintoosi et al.’s protocol [18] cannot withstand identity and password guessing attacks and smart card stolen attacks. Finally, Hajian et al.’s protocol [27] is vulnerable to device capture and session key disclosure attacks.

## 3. Preliminaries

This section introduces the PUFTAP-IoT system model and an adversary model for security analysis of authentication protocols. In addition, we briefly describe PUF, fuzzy extraction, and honey_list, which are the security technologies adopted in the proposed IoT-TFBAP.

### 3.1. The Proposed System Model

The system model of PUFTAP-IoT is shown in Figure 1. PUFTAP-IoT consists of following three entities:**User**: The user requests communication to the gateway to use the sensing device. Only registered users can use IoT services by requesting communication to the gateway.**Sensing device**: Sensing devices are smart devices deployed in various IoT environments. Examples in Figure 1 include smart agriculture, vehicles, smart doors, and smart watches. They collect data and provide it to users, and users can use the data to execute any commands they want. Sensing devices also have limited computational power.**Gateway**: All service users and sensing devices must be registered with the gateway. A gateway is a trusted entity that is responsible for the process and regulates authentication requests between users and sensing devices.

Users must first register with the gateway when they want to communicate with a sensing device. The gateway stores relevant data from users and sensing devices, and controls communication between users and sensing devices. PUFTAP-IoT consists of a registration phase, login and authentication phase, and password and biometrics update phase. In the registration phase, users and sensing devices are registered with the gateway through secure channels. During the login and authentication phase, the user, gateway, and sensing device authenticate each other and generate a session key for communication. In the future, the user can safely communicate with the sensing device using this session key. In the password and biometrics update phase, users can update their passwords and biometrics if desired. To defend against malicious adversaries’ ID/password guessing attacks and brute-force attacks, the gateway creates and stores honey_list. In addition, the sensing devices have built-in PUF technology to protect them from physical capture attacks.

### 3.2. The Adversary Model

We adopt the “Dolev-Yao (DY) adversary model” [36] to analyze the proposed protocols [15,18,27] and the IoT-TFBAP. The DY adversary model is a widely adopted model to analyze the security of wireless networks and assumes the following:The adversary can learn messages by intercepting messages delivered over insecure, public wireless channels. Through the learned message, the adversary can create a valid message and insert and modify it.The adversary can obtain stored values by stealing a valid user’s smart card and sensing device [37].The adversary can guess the user’s ID/password pair in polynomial time [21].The adversary can perform guessing, impersonation, known session-specific temporary information, and session key disclosure attacks using the acquired values.

### 3.3. Physical Unclonable Function

We adopt PUF technology to securely store secret parameters in the sensing device. PUF can be described as “the representation of the unique, non-replicable, instance-specific functionality of a physical entity” [29]. The randomness and uncertainty in integrated circuit fabrication is less likely to create duplicates, making PUFs increasingly visible in the security realm. PUF receives the challenge *C* and obtains its response *R* through the physical properties of *C* and the integrated chip (IC). Since both the accepted *C* and the generated *R* are strings of bits, PUF is expressed as R=PUF(C) and can be considered as a one-way function. In an ideal situation, a one-to-one correspondence exists between a challenge–response pair and a PUF, where if a challenge is assigned to the same PUF multiple times, the generated response is the same, and when the same challenge is given to different PUFs, the response obtained is different. PUF also has the following characteristics:It is impossible to clone PUF to create the same device [38].Any attempt to change the device containing the PUF will change the PUF’s behavior and destroy the PUF [39].In real-world manufacturing circuits, the difference between mapping input and output functions is fixed and unpredictable. In this respect, the hardware is equivalent to a one-way function [40].

However, due to environmental and circuit noise, PUFs always output varying responses with some margin of error in *C*s. To solve this problem, PUF is being applied with fuzzy extractor technology [41].

### 3.4. Fuzzy Extraction

To solve the problem of noisy PUF, we introduce fuzzy extraction technology [41]. Moreover, we can use fuzzy extraction to solve the noise that can occur in the biometric input. The fuzzy extractor consists of the GEN function and the REP function.

The GEN function is for generating key information corresponding to the entered value. Entering the data $D_{i}$ into the GEN function outputs the secret key data $R_{i}$, which is a uniform random string. The GEN function also outputs the string $P_{i}$, which helps to remove the noise and recover the key value.

The REP function restores the secret key $R_{i}$. Enter the data $D_{i}$ and the helper string $P_{i}$ into the REP function. At this time, $D_{i}$ may generate noise. For this, $P_{i}$ helps to output the correct $R_{i}$. To recover the same $R_{i}$, the metric space distance between $D_{i}$ and $D_{i}^{{}^{\prime }}$ must be within the specified tolerance.

### 3.5. Honey List

Assume that attackers attempt to obtain useful data by performing brute-force and online-guessing attacks. In this case, honey_list prevents the algorithm “Honey Encryption (HE)” [30,31] from attempting to obtain data by guessing the password. If an adversary attempts attacks with the wrong password, HE uses an algorithm to generate fake valid messages, “Honey words”. [42] has more details on the honey word generation algorithm.

Various methods have been used to resist brute-force or online-guessing attacks using honey_list at the login and authentication phase. Out of all of them, PUFTAP-IoT calls honey_list by adopting the following method. If an attacker tries to login using the guessing password, the login proceeds as usual, but the gateway monitors the attacker’s login source for intrusion detection. The gateway also kills the session “when the number of entries in honey_list exceeds a predefined threshold” and notifies the user to update their password.

## 4. Cryptanalysis of Authentication Protocols

This section shows the analysis of various authentication protocols using sensor or sensing devices in an IoT environment. A review of each protocol is omitted, and for convenience of explanation, *S* (sensor) of Chunka et al. and Amintoosi et al. and *S* (sensing device) of Hajian et al. are all denoted as SD (sensing device). The rest of the notation is the same as that of each authentication protocol. Table 1 shows the notations used in this paper.

### 4.1. Cryptanalysis of Chunka et al.’s Protocol

We prove that Chunka et al.’s protocol [16] is not safe against known session-specific temporary information attacks and does not provide perfect forward secrecy.

#### 4.1.1. Known Session-Specific Temporary Information Attacks

Suppose that the adversary Adv obtains a session-specific temporary information $r_{1}$. Then, Adv is able to compute the legitimate session key. The detailed steps are as follows:

**Step 1: **Adv computes $h\left(\alpha_{i}⊕k\right)=r_{1}⊕MID_{i}$, since $MID_{i}$ is public parameter. Then, Adv can obtain $P_{i}$, where $P_{i}$ is obtained through an insecure channel.

**Step 2: **Adv computes $h(P_{i}||h\left(\alpha_{i}⊕k\right)\left|\right|r_{1})=E_{j}⊕h\left(r_{1}⊕r_{2}⊕r_{3}\right)$ via $E_{j}$, which is transmitted to the public channel.

**Step 3:** Finally, Adv can compute the legitimate session key $SK=h\left(r_{1}⊕r_{2}⊕r_{3}\right)||h(P_{i}||h\left(\alpha_{i}⊕k\right)\left|\right|r_{1})$.

#### 4.1.2. Off-Line Guessing Attacks

According to the adversary model in Section 3.2, the adversary Adv can guess the ID/PW pair in polynomial time. The detailed steps are as follows:

**Step 1: **Adv is able to obtain values $\left\{X_{i},Z_{i},Q_{i},R_{i},h\left(\cdot \right)\right\}$ stored on the smart card via smart card stolen attacks. Then, Adv picks $ID_{a}$/$PW_{a}$ and computes $r_{a}=X_{i}⊕h(ID_{a}||PW_{a})$.

**Step 2: **Adv calculates $Z_{i_{a}}=h(ID_{a}||PW_{a}\left|\right|r_{a})$ and checks if $Z_{i_{a}}=Z_{i}$.

**Step 3:** If they are the same, Adv has successfully guessed the correct ID/password pair for the user. Otherwise, Adv repeats Steps 1 and 2.

#### 4.1.3. Impersonation Attacks

After off-line guessing attacks, Adv can impersonate the valid user. The detailed steps are as follows.

**Step 1:** Through guessing attacks, Adv computes $h(ID_{i}||r)$. Then, Adv can compute $h\left(\alpha_{i}\right)=Q_{i}⊕h(ID_{i}\left||r\right)$ and $h\left(\alpha_{i}⊕k\right)=R_{i}⊕h(ID_{i}\left||r\right)$ because $R_{i}$ and $Q_{i}$ are values stored in the smart card.

**Step 2:** Then, Adv generates a random nonce $r_{1_{a}}$ and computes $Mid_{a}=h\left(\alpha_{i}⊕k\right)⊕r_{a}$, $N_{a}=h(h\left(\alpha_{i}⊕k\right)||h\left(\alpha_{i}\right)\left|\right|r_{a})$.

**Step 3:** Finally, Adv sends the message $\left\{P_{i},MID_{i},N_{i}\right\}$. Thus, Adv can impersonate the legitimate user.

### 4.2. Cryptanalysis of Amintoosi et al.’s Protocol

This section shows that Amintoosi et al.’s protocol [18] is not secure to smart card stolen, off-line guessing, and impersonation attacks.

#### 4.2.1. Off-line Guessing Attacks

The adversary Adv can obtain the sensitive information stored in the smart card. Then, Adv can guess the ID/password pair in polynomial time. The detailed steps are as follows:

**Step 1: **Adv can obtain values $\left\{b_{i},A_{i},B_{i},a_{i}\right\}$ stored on the smart card. Then, Adv picks $ID_{a}$/$PW_{a}$ and computes $p_{a}=h(ID_{a}||PW_{a}\left|\right|a_{i})$, $N_{a}=A_{a}⊕p_{a}$, and $bID_{a}=h(b_{i}||ID_{a})$.

**Step 2: **Adv calculates $M_{a}=h(b_{i}||ID_{a}||bID_{a})$ and $B_{a}=h(M_{a}||ID_{a}||bID_{a})$. Then, Adv checks if $B_{a}=B_{i}$.

**Step 3:** If they are the same, Adv has successfully guessed the correct ID/password pair for the user. Otherwise, Adv repeats Steps 1 and 2.

#### 4.2.2. Impersonation Attacks

After guessing the legitimate user’s ID/password pair, the Adv computes $\{M1_{i},M2_{i},T_{1},b_{i},$$M2_{i}\}$ can be masquerading. The detailed steps are as following.

**Step 1:** After off-line guessing attacks, Adv obtains valid values $M_{i}$ and $N_{i}$. Then, Adv can compute $d_{i}=M1_{i}⊕h(M_{i}\left|\right|N_{i}\left|\right|T_{1})$ to obtain $d_{i}$, where $M1_{i}$ is transmitted to the public channel.

**Step 2:** Then, Adv also can compute $M2_{i}=h(M_{i}\left|\right|N_{i}\left|\right|d_{i})$.

**Step 3:** Therefore, Adv can compute $M1_{i},T_{1},b_{i},$ and $M2_{i}$. This means that Adv can impersonate the valid user. So, we can say that Amintoosi et al.’s protocol is not secure against impersonation attacks.

### 4.3. Cryptanalysis of Hajian et al.’s Protocol

In this section, we show that Hajian et al.’s protocol [27] is vulnerable to device capture attacks, device impersonation attacks, and session key disclosure attacks.

#### 4.3.1. Device Impersonation Attacks

The adversary Adv can obtain the $\left\{SID_{j},x_{j},f_{j}\right\}$ stored in SD through a device capture attack. After that, Adv can impersonate as a valid SD by generating a message using the obtained values. After the device capture attacks, the detailed steps of the Adv’s device impersonation attack are as follows:

**Step 1: **Adv obtains the values $\left\{M_{2},T_{1}\right\}$ via the message sent to the public channel. Then, Adv can obtain $K_{j}$ through computing $K_{j}=h(x_{j}\left|\right|f_{j}\left|\right|T_{1})⊕M_{2}$.

**Step 2: **Adv can compute the legitimate $V_{2}=h(SID_{j}||(x_{j}\left|\right|⊕f_{j})||K_{j}\left|\right|T_{1})$. Finally, Adv can compute the valid response message $\left\{M_{2},V_{2}\right\}$. Thus, we can say that Adv can conduct device impersonation attacks.

#### 4.3.2. Session Key Disclosure Attacks

After Adv conducts device impersonation attacks, Adv obtains $x_{j},f_{j},$ and $K_{j}$. Adv can calculate the session key using these values. Therefore, an attacker can perform session key disclosure attacks, and the detailed steps are as follows:

**Step 1: **Adv can learn values $M_{1}$ and $M_{5}$ through the message sent over the open channel. Then, Adv can compute $M_{5}⊕h(T_{1}\left|\right|f_{j}⊕x_{j})⊕M_{1}=(K_{i}^{{}^{\prime }}||TID_{i}^{new})$.

**Step 2:** Then, Adv can obtain $K_{i}^{{}^{\prime }}$ and $TID_{i}^{new}$.

**Step 3:** Therefore, Adv can compute the session key $SK_{ij}=h(K_{i}^{{}^{\prime }}⊕K_{j}||SID_{j}||TID_{i}^{new})$. Thus, we can say that Hajian et al.’s protocol is not secure against session key disclosure attacks.

## 5. The Proposed PUFTAP-IoT

In this section, we describe the proposed PUFTAP-IoT. In the proposed protocol, we adopt PUF technology to withstand device capture attacks. Additionally, we also apply the user’s biometrics and honey_list to prevent online-guessing and brute-force attacks. Accordingly, our protocol is observed to be secure against various attacks. Finally, we propose a lightweight protocol using XOR and hash functions to consider the resource limitations of sensing devices and to prevent system down.

### 5.1. Registration Phase

In order for a service user to use IoT services through a sensing device in an IoT environment, first, he/she must register his/her information in the gateway. Moreover, the sensing device also registers its information in the gateway. The registration phase for service users and sensing devices is shown in Figure 2, and the detailed registration phase is described below.

#### 5.1.1. Service User Registration Phase

Service users create their own information through ID, password, and biometric information and register it with the gateway, and the gateway issues a smart card. Here are the detailed steps:

**Step 1:** The service user $U_{i}$ inputs his/her identity ID, password $PW_{i}$, and imprints his/her biometrics $B_{i}$. Then, $U_{i}$ generates α and $R_{u}$ and computes $Gen\left(B_{i}\right)=\left(R_{i},P_{i}\right)$, $HID_{i}=h(ID_{i}\left|\right|R_{i})$, and $HPW_{i}=h(ID_{i}||PW_{i}\left|\right|R_{u}\left|\right|R_{i})$. $U_{i}$ sends $\langle HID_{i},$$HPW_{i}⊕\alpha \rangle$ to the gateway GW through a secure channel.

**Step 2:** After receiving the registration request message, GW checks the uniqueness of $HID_{i}$. If it has the uniqueness, GW generates a random nonce $R_{gw}$ and GW computes $A_{i}=h(HID_{i}\left|\right|K_{gw}\left|\right|R_{gw})$, $B_{i}=A_{i}⊕\left(HPW_{i}⊕\alpha \right)$, and $C_{i}=h(A_{i}||HID_{i})$. Then, GW generates the temporary service user’s identity $THID_{i}$ and stores $\left\{\left(HID_{i},THID_{i}\right),R_{gw},honey\_list=null\right\}$ in its secure database. GW issues the smart card $SC=\langle B_{i},C_{i},THID_{i}\rangle$ to $U_{i}$ via a secure channel.

**Step 3: **$U_{i}$ computes $L_{i}=h(ID_{i}||PW_{i}\left|\right|R_{i})⊕R_{u}$, $B_{i}^{{}^{\prime }}=B_{i}⊕\alpha =A_{i}⊕HPW_{i}$, and $C_{i}^{{}^{\prime }}=h(C_{i}||HPW_{i})$. Then, $U_{i}$ deletes $B_{i}$ and $C_{i}$ in SC and stores $L_{i},B_{i}^{{}^{\prime }},$ and $C_{i}^{{}^{\prime }}$ in SC.

#### 5.1.2. Sensing Device Registration Phase

The sensing device $SD_{j}$ utilizes the PUF function for registration and registers its own information with GW. The detailed registration steps are as follows:

**Step 1: **$SD_{j}$ picks its identity $SID_{j}$ and PUF’s challenge $C_{j}$. $SD_{j}$ generates a random nonce $R_{sd}$ and computes $Req_{j}=SID_{j}⊕h\left(R_{sd}\right)$, $R_{j}=PUF\left(C_{j}\right)$, $Gen\left(R_{j}\right)=\langle SDR_{j},SDP_{j}\rangle$, and $HSID_{j}=h(SID_{j}||SDR_{j})$. After that, $SD_{j}$ transmits $\langle Req_{j},R_{sd},HSID_{j},C_{j}\rangle$ to GW through a closed channel.

**Step 2: **GW computes $SID_{j}=Req_{j}⊕h\left(R_{sd}\right)$ and GW generates a random secret key $RK_{j}$. GW also computes $PSID_{j}=h(HSID_{j}||RK_{j})$ and $SI_{j}=h(PSID_{j}||h(K_{gw}||RK_{j})$. Finally, GW stores $\left\{\left(HSID_{j},PSID_{j}\right),RK_{j},C_{j}\right\}$ in its database and transmits $\langle PSID_{j},SI_{j}\rangle$ to $SD_{j}$ through a closed channels.

**Step 3:** After receiving the message, $SD_{j}$ stores $\left\{SID_{j},PSID_{j},SI_{j},SDP_{j}\right\}$.

### 5.2. Login and Authentication Phase

$U_{i}$ sends an authentication request message to GW after login through his/her smart card and credential information. After confirming this, GW sends an authentication message to the corresponding $SD_{j}$, and each entity authenticates the response message. When authentication is completed, $U_{i}$, GW, and $SD_{j}$ agree on a session key Skey, and secure communication can be guaranteed later through Skey. In addition, $U_{i}$ and GW update $THID_{i}$ to $THID_{inew}$ when authentication and key agreement are successful. The detailed formula is as follows, and the entire steps are summarized in Figure 3:

**Step 1:** The service user $U_{i}$ inserts SC and inputs $ID_{i}$, $PW_{i}$, and $B_{i}$. Then, SC computes $Rep\left(B_{i},P_{i}\right)=R_{i}$, $HID_{i}=h(ID_{i}\left|\right|R_{i})$, $R_{u}=L_{i}⊕h(ID_{i}||PW_{i}\left|\right|R_{i})$, $HPW_{i}=h(ID_{i}$$||PW_{i}\left|\right|$$R_{u}\left|\right|$$R_{i})$, $A_{i}=B_{i}^{{}^{\prime }}⊕HPW_{i}$, and $C_{i}^{*}=h(h(A_{i}||HID_{i}||HPW_{i})$. SC checks $C_{i}=C_{i}^{*}$. If it is correct, SC generates a random nonce $N_{u}$ and timestamp $T_{1}$. After that, SC computes $Msg_{1}=h(h(N_{u}\left|\right|A_{i})||A_{i}||HID_{i}||PSID_{j})$, $V_{1}=h(N_{u}\left|\right|A_{i})⊕h(HID_{i}\left|\right|A_{i}\left|\right|T_{1})$. $U_{i}$ sends the message $\langle Msg_{1},V_{1},T_{1},THID_{i},PSID_{j}\rangle$ through an open channel.

**Step 2:** When GW receives the request message, GW checks $|T_{1}-T_{1}^{*}|<\Delta T$. GW retrieves $HID_{i}$ corresponding to $THID_{i}$ and GW computes $A_{i}=h(HID_{i}\left|\right|K_{gw}\left|\right|R_{gw})$, $h(N_{u}\left|\right|A_{i})=h(HID_{i}\left|\right|A_{i}\left|\right|T_{1})⊕V_{1}$, $Msg_{1}^{*}=h(h(N_{u}\left|\right|A_{i})||A_{i}||HID_{i}||PSID_{j})$. Then, GW checks if $Msg_{2}=Msg_{2}^{*}$. If it is not same, GW inserts $A_{i}^{*}$ into honey_list. Otherwise, GW retrieves $\left(C_{j},RK_{j}\right)$ corresponding to $PSID_{j}$. GW generates a random nonce $N_{g}$ and timestamp $T_{2}$ and computes $SI_{j}=h(PSID_{j}||h(K_{gw}||RK_{j}))$, $V_{2}=C_{j}⊕h(PSID_{j}||SI_{j})$, $V_{3}=h(h(N_{u}\left|\right|A_{i})||h(N_{g}||SI_{j}))⊕h(HSID_{j}\left|\right|C_{j}||SI_{j})$, and $Msg_{2}=h(h(h(N_{u}\left|\right|A_{i})||h(N_{g}||SI_{j}))$$\left|\right|T_{2}\left|\right|$$HSID_{j}\left|\right|C_{j}||SI_{j})$. After computing, GW sends $\langle Msg_{2},V_{2},V_{3},T_{2})\rangle$ to $SD_{j}$ via an open wireless channel.

**Step 3: **$SD_{j}$ checks $|T_{2}-T_{2}^{*}|<\Delta T$. Then, $SD_{j}$ computes $C_{j}=V_{2}⊕h(PSID_{j}||SI_{j})$, $PUF\left(C_{j}\right)=R_{j}$, $Rep\left(R_{j},SDP_{j}\right)=SDR_{j}$, $HSID_{j}=h(SID_{j}||SDR_{j})$, $K_{gs}(=h(h(N_{u}\left|\right|A_{i})||$$h(N_{g}$$||SI_{j})))=V_{3}⊕h(HSID_{j}\left|\right|C_{j}||SI_{j})$, and $Msg_{2}^{*}=h(K_{GS}\left|\right|T_{2}||HSID_{j}\left|\right|C_{j}||SI_{j})$. Then, $SD_{j}$ checks $Msg_{2}=Msg_{2}^{*}$. If it is the same, $SD_{j}$ generates a random nonce $N_{sd}$ and timestamp $T_{3}$ and $SD_{j}$ computes a session key $Skey=h(N_{sd}||K_{gs})$. $SD_{j}$ also computes $V_{4}=Skey⊕h(HSID_{j}||SI_{j}\left|\right|C_{j}\left|\right|T_{3})$ and $Msg_{3}=h(C_{j}||HSID_{j}\left||Skey\right)$. After that, $SD_{j}$ sends the response message $\langle Msg_{3},V_{4},T_{3}\rangle$ to GW.

**Step 4: **GW computes the session key $Skey=V_{4}⊕(HSID_{j}||SI_{j}\left|\right|C_{j}\left|\right|T_{3})$, and computes $Msg_{3}^{*}=h(C_{j}||HSID_{j}\left||Skey\right)$. Then, GW checks if $Msg_{3}=Msg_{3}^{*}$. If it verifies, GW computes $THID_{inew}=h(h(N_{u}\left|\right|A_{i})||N_{g}||THID_{i})$, $V_{5}=Skey⊕h(h(N_{u}\left|\right|A_{i})||HID_{i})$, $V_{6}=THID_{inew}⊕h(HID_{i}||THID_{i}||h(N_{u}\left|\right|A_{i}))$, and $Msg_{4}=h(Skey||THID_{inew})$. After computing, GW transmits $\langle Msg_{4},$$V_{5},V_{6}\rangle$ to $U_{i}$ through an insecure channel.

**Step 5:** After receiving the response message, $U_{i}$ computes a session key $Skey=V_{5}⊕h(h(N_{u}\left|\right|A_{i})||HID_{i})$. Additionally, $U_{i}$ also computes $V_{6}=THID_{inew}⊕h(HID_{i}||THID_{i}||h(N_{u}\left|\right|A_{i}))$ and $Msg_{4}^{*}=h(Skey||THID_{inew})$. Then, $U_{i}$ checks if $Msg_{4}=Msg_{4}^{*}$. If it is correct, the session key is authentic, and $U_{i}$ and GW update $THID_{inew}$.

### 5.3. Service User Password and Biometrics Update Phase

Assume that the service user $U_{i}$ wants to use SC to change to a new password and biometrics. Specifically, this phase runs locally without any additional connections to GW, reducing computation and communication overhead. The following steps are the password and biometrics update process:

**Step 1:** The service user $U_{i}$ inputs his/her identity ID and password $PW_{i}$ and imprints his/her biometrics $B_{i}$. Then, SC computes $Rep\left(B_{i},P_{i}\right)=R_{i}$, $HID_{i}=h(ID_{i}\left|\right|R_{i})$, $R_{u}=L_{i}⊕h(ID_{i}||PW_{i}\left|\right|R_{i})$, $HPW_{i}=h(ID_{i}$$||PW_{i}\left|\right|$$R_{u}\left|\right|$$R_{i})$, $A_{i}=B_{i}^{{}^{\prime }}⊕HPW_{i}$, and
$C_{i}^{*}=h(h(A_{i}||HID_{i}||HPW_{i})$. SC checks $C_{i}=C_{i}^{*}$. If it is valid, SC asks $U_{i}$ to enter the new password and biometrics.

**Step 2: **$U_{i}$ enters a new password $PW_{inew}$ and new biometrics $B_{inew}$. SC proceeds to compute parameters $GEN\left(B_{inew}\right)=\left(R_{inew},P_{inew}\right)$, $HPW_{inew}=h(ID_{i}||PW_{inew}\left|\right|R_{u}\left|\right|R_{inew})$, $L_{inew}=h(ID_{i}||PW_{inew}\left|\right|R_{inew})⊕R_{u}$, $B_{inew}^{{}^{\prime }}=B_{i}^{{}^{\prime }}⊕HPW_{i}⊕HPW_{inew}$, and $C_{inew}^{{}^{\prime }}=h(C_{i}^{{}^{\prime }}||HPW_{i})$. Then, SC replaces $L_{i},B_{i}^{{}^{\prime }},$ and $C_{i}^{{}^{\prime }}$ with $L_{inew},B_{inew}^{{}^{\prime }},$ and $C_{inew}^{{}^{\prime }}$.

## 6. Security Analysis

In this section, we analyze the security of PUFTAP-IoT. We first show that the protocol is safe against various attacks through informal analysis. In addition, we prove that mutual authentication and session key agreement of the protocol can be safely achieved through the universally used BAN logic and ROR model. Finally, we demonstrate the security of PUFTAP-IoT on a wireless network using the scyther simulation tool.

### 6.1. Informal Security Analysis

Here, we perform an informal (non-mathematical) security analysis to show that PUFTAP-IoT is safe against various attacks and also provides various security features.

#### 6.1.1. Offline and Online-Guessing Attacks

Assume that the adversary Adv obtains the $U_{i}$’s SC and attempts an offline-guessing attack using parameters $\left\{L_{i},B_{i}^{{}^{\prime }},C_{i}^{{}^{\prime }},THID_{i}\right\}$ in SC. However, since Adv is the value that $R_{i}$ should be calculated as, the biometric of $U_{i}$, $R_{u}=L_{i}⊕h(ID_{i}||PW_{i}\left|\right|R_{i})$ could not be calculated. Moreover, Adv tries an online-guessing attack for obtaining $U_{i}$’s sensitive information. Unfortunately, the attacker does not know if the correct ID and password were guessed because of the honey_list stored on the gateway system. Moreover, PUFTAP-IoT is safe from online-guessing attacks because the session is terminated when the threshold of honey_list is exceeded. Therefore, PUFTAP-IoT is safe against offline- and online-guessing attacks.

#### 6.1.2. Service User Anonymity

If Adv steals $U_{i}$’s SC and obtains values stored in SC, Adv tries to obtain $U_{i}$’s real identity, pseudo-identity or temporary identity. However, Adv cannot obtain the ID of $U_{i}$ and $HID_{i}$ because $HID_{i}$ is masked by the hash function and $R_{i}$. Although $THID_{i}$ is transmitted through the public channel, $THID_{inew}$ is updated by GW when authentication and key agreement are successful. In addition, $THID_{inew}$ is masked with $N_{u}$ and $N_{g}$, and these values change every session. Therefore, PUFTAP-IoT can safely guarantee the anonymity of service users.

#### 6.1.3. Impersonation Attack

In order for Adv to disguise $U_{i}$, GW, and $SD_{j}$, Adv must be able to compute the messages sent to the public channel. Messages sent from PUFTAP-IoT to public channels change per session due to random values $N_{u}$, $N_{s}$, and $N_{sd}$ and timestamps. In addition, $THID_{i}$ is also updated to $THID_{inew}$ when the authentication is successful, so Adv cannot calculate the correct message. Therefore, PUFTAP-IoT is resistant to impersonation attacks.

#### 6.1.4. Sensing Device Physical Capture Attack

When Adv performs a physical capture attack on $SD_{j}$, Adv can obtain $\{SID_{j},PSID_{j},SI_{j},$$SDP_{j}\}$ stored in $SD_{j}$. However, Adv cannot calculate the correct session key through these parameters. In order for Adv to calculate the session key, $K_{gs}(=h(h(N_{u}\left|\right|A_{i})||h(N_{g}$$||SI_{j})))=V_{3}⊕h(HSID_{j}\left|\right|C_{j}||SI_{j})$ must be calculated. However, since Adv cannot obtain $R_{j}$, Adv cannot compute $SDR_{j}$. Therefore, Adv is not able to compute $HSID_{j}=h(SID_{j}||SDR_{j})$. This is because $R_{j}$ is a value created by the PUF function, and the PUF is a function that is a physically unclonable circuit and cannot be duplicated. Therefore, PUFTAP-IoT is safe against sensing device physical capture attacks.

#### 6.1.5. Replay and Man-in-the-Middle Attack

We assume that Adv obtains messages transmitted over a public channel and information of $U_{i}$’s SC and $SD_{j}$. However, Adv cannot compute $U_{i}$’s valid message as mentioned in Section 6.1.3. Additionally, Adv also cannot generate $SD_{j}$’s valid messages according to Section 6.1.4. Additionally, every message changes with $N_{u},N_{g},andN_{sd}$ and timestamps every session. Therefore, we can say that PUFTAP-IoT is secure against replay and man-in-the-middle attacks.

#### 6.1.6. Stolen Verifier Attack

Suppose Adv obtains the GW verification tables $\left\{HID_{i},THID_{i},R_{gw},honey\_list=null\right\}$ and $\left\{HSID_{j},PSID_{j},RK_{j},C_{j}\right\}$ to compute the session key Skey or perform impersonation attacks. However, Adv cannot compute $A_{i}=h(HID_{i}\left|\right|K_{gw}\left|\right|R_{gw})$ and $SI_{j}=h(PSID_{j}||h(K_{gw}||RK_{j})$ without GW’s secret key $K_{gw}$. Furthermore, due to the nature of the PUF function, Adv cannot compute $R_{j}=PUF\left(C_{j}\right)$. Therefore, Adv cannot perform session key and impersonation attacks. Accordingly, we can say that PUFTAP-IoT is resistant to stolen verifier attacks.

#### 6.1.7. Perfect Forward Secrecy

Assuming that GW’s secret key $K_{gw}$, is leaked to Adv, Adv can try to calculate Skey through $K_{gw}$. However, since $A_{i}$ and $SI_{j}$ are masked with $K_{gw}$ as well as $R_{gw}$ and $RK_{j}$ which are secret keys generated for each entity, Adv cannot compute $A_{i}$ and $SI_{j}$. Therefore, since Adv cannot calculate valid Skey, PUFTAP-IoT can guarantee perfect forward secrecy.

#### 6.1.8. Session-Specific Random Number Leakage Attack

Assume that $N_{u},N_{g},andN_{sd}$, which are random nonces generated in the session, were leaked to Adv. With these values, Adv will try to compute Skey. However, Adv cannot compute the session key $Skey=h(N_{sd}||h(h(N_{u}\left|\right|A_{i})||h(N_{g}||SI_{j})))$. To calculate a valid Skey, $A_{i}$ and $SI_{j}$ must be calculated, but as mentioned in the Sections above, $A_{i}$ and $SI_{j}$ cannot be calculated by Adv. Therefore, PUFTAP-IoT is safe against session-specific random number leakage attacks.

#### 6.1.9. Session Key Disclosure Attack

Adv wants to compute the Skey for obtaining sensitive information. However, as discussed in Section 6.1.6, Section 6.1.7 and Section 6.1.8, Adv cannot compute the valid Skey because of the computationally infeasible problem. Thus, PUFTAP-IoT prevents session key disclosure attacks.

#### 6.1.10. Mutual Authentication

In PUFTAP-IoT, all entities authenticate each other by verifying messages containing $Msg_{1},Msg_{2},$$Msg_{3},$ and $Msg_{4}$. Moreover, these messages are changed with random numbers and current timestamps. After all entities authenticate each other, they compute the same Skey. Thus, PUFTAP-IoT guarantees mutual authentication.

### 6.2. BAN Logic

For proving that PUFTAP-IoT is able to provide secure authentication, we conduct BAN logic [32]. The notations used in BAN logic are shown in Table 2, and the five rules used are as follows [43,44,45]:**1.** 
Jurisdiction rule:
$$\frac{\chi \;\left|\equiv \omega \;\right|⟹\varepsilon ,\;\chi \;\left|\equiv \omega \;\right|\equiv \varepsilon }{\chi \;|\equiv \varepsilon }$$**2.** 
Message meaning rule:
$$\frac{\chi \;|\equiv \chi \overset{K}{↔}\omega ,\;\chi ⊲{\left\{\varepsilon \right\}}_{K}}{\chi \;\left|\equiv B\;\right|∼\varepsilon }$$**3.** 
Nonce verification rule:
$$\frac{\chi \;\left|\equiv \#(\varepsilon ),\;\chi \;\right|\equiv \omega \;|∼\varepsilon }{\chi \;\left|\equiv \omega \;\right|\equiv \varepsilon }$$**4.** 
Belief rule:
$$\frac{\chi \;|\equiv \left(\varepsilon ,F\right)}{\chi \;|\equiv \varepsilon }$$**5.** 
Freshness rule:
$$\frac{\chi \;|\equiv \#(\varepsilon )}{\chi \;|\equiv \#\left(\varepsilon ,F\right)}$$

To implement BAN logic, we describe logical rules, goals, assumptions, and ideal forms, thereby proving that PUFTAP-IoT provides secure mutual authentication.

#### 6.2.1. Goals

In order to prove that secure mutual authentication is achieved, the following goals must be achieved:**Goal 1:** $U_{i}|\equiv \left(U_{i}\overset{Skey}{⟷}GW\right)$**Goal 2:** $U_{i}\left|\equiv GW\right|\equiv \left(U_{i}\overset{Skey}{⟷}GW\right)$**Goal 3:** $GW|\equiv (U_{i}\overset{Skey}{⟷}GW)$**Goal 4:** $GW|\equiv U_{i}|\equiv \left(U_{i}\overset{Skey}{⟷}GW\right)$**Goal 5:** $SD_{j}|\equiv \left(SD_{j}\overset{Skey}{⟷}GW\right)$**Goal 6:** $SD_{j}\left|\equiv GW\right|\equiv \left(SD_{j}\overset{Skey}{⟷}GW\right)$**Goal 7:** $GW|\equiv (SD_{j}\overset{Skey}{⟷}GW)$**Goal 8:** $GW|\equiv SD_{j}|\equiv \left(SD_{j}\overset{Skey}{⟷}GW\right)$

#### 6.2.2. Idealized Forms

The idealized forms are:$M_{1}$:$\;U_{i}\to GW:\{h(N_{u}\left|\right|A_{i}{)\}}_{HID_{i}}$$M_{2}$:$\;GW\to SD_{j}:\{h(h(N_{u}\left|\right|A_{i})||h(N_{g}||SI_{j}{))\}}_{SI_{j}}$$M_{3}$:$\;SD_{j}\to GW:\{h(N_{sd}\left|\right|K_{gs}{)\}}_{SI_{j}}$$M_{4}$:$\;GW\to U_{i}:\{h(N_{sd}\left|\right|K_{gs}{)\}}_{HID_{i}}$

#### 6.2.3. Assumptions

The following assumptions are generated for the initial state of PUFTAP-IoT to achieve the BAN logic proof:$A_{1}$:$\;GW|\equiv U_{i}\overset{HID_{i}}{⟷}GW$$A_{2}$:$\;GW|\equiv \#(N_{u})$$A_{3}$:$\;SD_{j}|\equiv GW\overset{SI_{j}}{⟷}SD_{j}$$A_{4}$:$\;SD_{j}|\equiv \#\left(N_{g}\right)$$A_{5}$:$\;GW|\equiv GW\overset{SI_{j}}{⟷}SD_{j}$$A_{6}$:$\;GW|\equiv \#(N_{sd})$$A_{7}$:$\;U_{i}|\equiv U_{i}\overset{HID_{i}}{⟷}GW$$A_{8}$:$\;U_{i}|\equiv \#\left(N_{g}\right)$$A_{9}$:$\;U_{i}\left|\equiv GW\right|\Rightarrow \left(U_{i}\overset{Skey}{⟷}GW\right)$$A_{10}$:$\;GW|\equiv U_{i}|\Rightarrow \left(U_{i}\overset{Skey}{⟷}GW\right)$$A_{11}$:$\;SD_{j}\left|\equiv GW\right|\Rightarrow \left(SD_{j}\overset{Skey}{⟷}GW\right)$$A_{12}$:$\;GW|\equiv SD_{j}|\Rightarrow \left(SD_{j}\overset{Skey}{⟷}GW\right)$

#### 6.2.4. Proof

The main proof using the rules and assumptions of BAN logic is:**Step 1**:$S_{1}$ can be obtained from $M_{1}$.
$$S_{1}:GW◃\{h(N_{u}\left|\right|A_{i}{)\}}_{HID_{i}}$$**Step 2**:$S_{2}$ can be obtained by applying the MMR with $A_{1}$.
$$S_{2}:GW|\equiv U_{i}|∼\{h(N_{u}\left|\right|A_{i}{)\}}_{HID_{i}}$$**Step 3**:$S_{3}$ can be gained from the FR with $S_{2}$ and $A_{2}$.
$$S_{3}:GW|\equiv \#(h(N_{u}\left|\right|A_{i}))$$**Step 4**:$S_{4}$ can be acquired by applying the NVR with $S_{2}$ and $S_{3}$.
$$S_{4}:GW|\equiv U_{i}|\equiv (h(N_{u}\left|\right|A_{i}))$$**Step 5**:$S_{5}$ can be obtained from $M_{2}$.
$$S_{5}:SD_{j}◃\{h(h(N_{u}\left|\right|A_{i})||h(N_{g}||SI_{j}{))\}}_{SI_{j}}$$**Step 6**:$S_{6}$ can be gained from MMR with $S_{5}$ and $A_{3}$.
$$S_{6}:SD_{j}|\equiv GW|∼\{h(h(N_{u}\left|\right|A_{i})||h(N_{g}||SI_{j}{))\}}_{SI_{j}}$$**Step 7**:$S_{7}$ can be obtained by applying FR with $S_{6}$ and $A_{4}$.
$$S_{7}:SD_{j}|\equiv \#(h(h(N_{u}\left|\right|A_{i})||h(N_{g}||SI_{j})))$$**Step 8**:$S_{8}$ can be obtained from NVR with $S_{6}$ and $S_{7}$.
$$S_{8}:SD_{j}|\equiv GW|\equiv (h(h(N_{u}\left|\right|A_{i})||h(N_{g}||SI_{j})))$$**Step 9**:$S_{9}$ can be obtained from $M_{3}$.
$$S_{9}:GW◃\{h(N_{sd}\left|\right|K_{gs}{)\}}_{SI_{j}}$$**Step 10**:$S_{10}$ can be gained from MMR with $S_{9}$ and $A_{5}$.
$$S_{10}:GW|\equiv SD_{j}|∼\{h(N_{sd}\left|\right|K_{gs}{)\}}_{SI_{j}}$$**Step 11**:$S_{11}$ can be obtained by applying FR with $S_{10}$ and $A_{6}$.
$$S_{11}:GW|\equiv \#(h(N_{sd}\left|\right|K_{gs}))$$**Step 12**:$S_{12}$ can be obtained from NVR with $S_{10}$ and $S_{11}$.
$$S_{12}:GW|\equiv SD_{j}|\equiv (h(N_{sd}\left|\right|K_{gs}))$$**Step 13**:$S_{13}$ can be obtained from $M_{4}$.
$$S_{13}:U_{i}◃\{h(N_{sd}\left|\right|K_{gs}{)\}}_{HID_{i}}$$**Step 14**:$S_{14}$ can be obtained from MMR with $S_{13}$ and $A_{7}$.
$$S_{14}:U_{i}|\equiv GW|∼\{h(N_{sd}\left|\right|K_{gs}{)\}}_{HID_{i}}$$**Step 15**:$S_{15}$ can be obtained from FR with $S_{14}$ and $A_{8}$, since $K_{gs}=h(h(N_{u}\left|\right|A_{i})||h(N_{g}||SI_{j}))$.
$$S_{15}:U_{i}|\equiv \#(h(N_{sd}\left|\right|K_{gs}))$$**Step 16**:$S_{16}$ can be obtained from NVR with $S_{14}$ and $S_{15}$.
$$S_{16}:U_{i}|\equiv GW|\equiv (h(N_{sd}\left|\right|K_{gs}))$$**Step 17**:$S_{17}$ and $S_{18}$ can be obtained from $S_{8}$ and $S_{12}$ since $Skey=h(N_{sd}||K_{gs})$.
$$S_{17}:SD_{j}\left|\equiv GW\right|\equiv \left(SD_{j}\overset{Skey}{⟷}GW\right)\;\left(\mathrm{Goal}6\right)$$
$$S_{18}:GW\left|\equiv SD_{j}\right|\equiv \left(SD_{j}\overset{Skey}{⟷}GW\right)\;\left(\mathrm{Goal}8\right)$$**Step 18**:$S_{19}$ and $S_{20}$ can be obtained by applying JR from $S_{17}$, $S_{18}$, $A_{11}$, and $A_{12}$.
$$S_{19}:SD_{j}|\equiv \left(SD_{j}\overset{Skey}{⟷}GW\right)\;\left(\mathrm{Goal}5\right)$$
$$S_{20}:GW|\equiv \left(SD_{j}\overset{Skey}{⟷}GW\right)\;\left(\mathrm{Goal}7\right)$$**Step 19**:$S_{21}$ and $S_{22}$ can be obtained from $S_{4}$ and $S_{16}$ since $Skey=h(N_{sd}||K_{gs})$.
$$S_{21}:U_{i}\left|\equiv GW\right|\equiv \left(U_{i}\overset{Skey}{⟷}GW\right)\;\left(\mathrm{Goal}2\right)$$
$$S_{22}:GW\left|\equiv U_{i}\right|\equiv \left(U_{i}\overset{Skey}{⟷}GW\right)\;\left(\mathrm{Goal}4\right)$$**Step 20**:$S_{23}$ and $S_{24}$ can be obtained by applying JR from $S_{21}$, $S_{22}$, $A_{9}$, and $A_{10}$.
$$S_{23}:U_{i}|\equiv \left(U_{i}\overset{Skey}{⟷}GW\right)\;\left(\mathrm{Goal}1\right)$$
$$S_{24}:GW|\equiv \left(U_{i}\overset{Skey}{⟷}GW\right)\;\left(\mathrm{Goal}3\right)$$

Therefore, we prove that PUFTAP-IoT can satisfy all goals of BAN logic. Accordingly, it can be said that PUFTAP-IoT can guarantee secure mutual authentication.

### 6.3. ROR

We use the ROR model [33] to describe the semantic security of PUFTAP-IoT. We demonstrate that session key security can be guaranteed through the ROR model [46,47,48]. This section briefly describes the ROR model and presents a proof of the protocol’s session key security in Theorem 1. PUFTAP-IoT in the ROR model has three participants $P^{t}$, which are service user $P_{U_{i}}^{t_{1}}$, gateway $\;mathcalP_{GW}^{t_{2}}$, and sensing device $P_{S_{j}}^{t_{3}}$. Additionally, for each participant, *t*th represents an instance of the running participant. We assume that an attacker Adv can modify, remove, insert or learn messages sent during communication. In the ROR model, various queries are defined to simulate real-world attacks, Execute, CorruptSC, Reveal, Send, and Test queries. A detailed description of the query follows:$Execute(P_{U_{i}}^{t_{1}},P_{GW}^{t_{2}},P_{SD_{j}}^{t_{3}})$: Adv conducts Execute query to obtain messages sent over insecure channels between $U_{i}$, GW, and $SD_{j}$.$CorruptSC(P_{U_{i}}^{t_{1}})$: CorruptSC indicates that Adv can obtain information stored in the smart card of $U_{i}$.$Reveal(P^{t})$: $Reveal(P^{t})$ is that Adv returns the session key Skey between $P_{U_{i}}^{t_{1}}$, $P_{GW}^{t_{2}}$, and $P_{SD_{j}}^{t_{3}}$. Skey is safe if Adv reveals Skey using the $Reveal(P^{t})$ query.$Send(P^{t},M)$: Send query allows Adv to transmit the *M* message to $P^{t}$ and receive a response.$Test(P^{t})$: A fair coin fc is tossed before the game starts, and the result is known only to Adv. Adv uses this result to determine Test query. If Adv conducts this query and Skey is fresh, $P^{t}$ will return Skey for fc = 1 or a random nonce for fc = 0. Otherwise, $P^{t}$ returns a null (⊥).

After Adv conducts Test query on participants, Adv has to separate resulting values. Adv checks the consistency of the random bit fc through the output of the Test query. For Adv to win the game, the guessed bit $fc^{\prime }$ must equal fc. Additionally, the collision-resistant one-way hash function h(·) is accessible to all participants. Model h(·) is a random oracle Hash.

#### Security Proof

**Theorem** **1.**
*Adv can obtain information by breaking the session key security. Mark the advantage of Adv running in polynomial time as $Adv_{t}$. Then, we obtain:*

$$Adv_{t}\leq \frac{q_{h}^{2}}{\left|Hash\right|}+\frac{q_{p}^{2}}{\left|PUF\right|}+2max\left\{C\cdot q_{s}^{s},\frac{q_{s}}{2^{l_{D}}}\right\}$$


*Here, $q_{h}$, $q_{p}$, and $q_{s}$ are the number of Hash, PUF, and Send queries, respectively. |Hash| and |PUF| are the range space of the hash function h(·) and PUF function PUF(·), respectively. In addition, C and s denote Zipf’s parameters, and $l_{D}$ is the number of bits in the biometric $B_{i}$ of $U_{i}$.*


**Proof.** We run four sequence games $GM_{i}$ to prove session key security, where i∈[0,4]. $Succ_{Adv,i}$ represents the event that Adv wins $GM_{i}$ by correctly guessing any bit fc. The advantage of Adv winning the game $GM_{i}$ is denoted by $Pr[Succ_{Adv,GM_{i}}]$. Each game is described below.
$GM_{0}$: Adv can execute a real attack on PUFTAP-IoT through this game. Adv selects fc at the beginning of $GM_{0}$. Then, according to this game, we obtain:
(1)$$Adv_{t}=\left|2Pr\left[Succ_{Adv,GM_{0}}\right]-1\right|$$$GM_{1}$: Adv conducts the $Execute(P_{U_{i}}^{t_{1}},P_{GW}^{t_{2}},P_{SD_{j}}^{t_{3}})$ query through this game and eavesdrops transmitted messages $<Msg_{1},V_{1},T_{1},THID_{i},PSID_{j}>$, $<Msg_{2},V_{2},V_{3},T_{2}>$, $<Msg_{3},V_{4}>$, and $<Msg_{4},V_{5},V_{6}>$. Adv then checks whether the derived Skey is real to execute Reveal and Test queries. In PUFTAP-IoT, the session key consists of $Skey=h(N_{sd}||K_{gs})$. To derive Skey, Adv must know the ID and random numbers of $U_{i}$, GW, and $SD_{j}$. As a result, Adv never increases the probability of winning $GM_{1}$. Thus, $GM_{0}$ and $GM_{1}$ can be considered indistinguishable, and we obtain:
(2)$$Pr\left[Succ_{Adv,GM_{1}}\right]=Pr\left[Succ_{Adv,GM_{0}}\right]$$$GM_{2}$: To obtain Skey, Adv conducts Hash and Send queries.Adv can modify exchanged messages to carry out active attacks. However, all exchanged messages are protected using the one-way hash function h(·) and consist of secret credentials and random numbers. Moreover, it is difficult for Adv to derive secret credentials and a random nonce because it is a computationally infeasible problem depending on the properties of h(·). So, using the birthday paradox, we obtain:
(3)$$|Pr\left[Succ_{Adv,GM_{2}}\right]-Pr\left[Succ_{Adv,GM_{1}}\right]|\leq \frac{q_{h}^{2}}{2|Hash|}$$$GM_{3}$: It is similar to $GM_{2}$. Adv conducts Send and PUF queries. As described in Section 3.3, PUF(·) has security properties. So, we can obtain the following result:
(4)$$|Pr\left[Succ_{Adv,GM_{3}}\right]-Pr\left[Succ_{Adv,GM_{2}}\right]|\leq \frac{q_{p}^{2}}{2|PUF|}$$$GM_{4}$: In $GM_{4}$, Adv can try to obtain Skey with the CorruptSC query. By the CorruptSC query, Adv can extract sensitive values $\left\{L_{i},B_{i}^{\prime },C_{i}^{\prime },THID_{i}\right\}$ stored on the smart card of $U_{i}$. $L_{i}$, $B_{i}^{\prime }$, and $C_{i}^{\prime }$ are expressed as $L_{i}=h(ID_{i}||PW_{i}\left|\right|R_{i})⊕R_{u}$, $B_{i}^{\prime }=B_{i}⊕\alpha =A_{i}⊕HPW_{i}$, and $C_{i}^{\prime }=h(C_{i}||HPW_{i})$. Since Adv has no knowledge of identity $ID_{i}$ and password $PW_{i}$, Adv must guess these parameters from the extracted values. However, it is computationally infeasible for Adv to guess ID, password, and $B_{i}$ simultaneously. Thus, $GM_{3}$ and $GM_{4}$ are indistinguishable. By utilizing Zipf’s law, we can obtain:
(5)$$|Pr\left[Succ_{Adv,GM_{4}}\right]-Pr\left[Succ_{Adv,GM_{3}}\right]|\leq max\left\{C\cdot q_{send}^{s},\frac{q_{s}}{2^{l_{D}}}\right\}$$Now that all the games were run, Adv has to guess the bit to win the game. Thus, we can obtain following results:
(6)$$Pr\left[Succ_{Adv,GM_{4}}\right]=\frac{1}{2}$$From Equations (1) and (2), we obtain the result as follows:
(7)$$\frac{1}{2}Adv_{t}=|Pr\left[Succ_{Adv,GM_{0}}-\frac{1}{2}\right]|=|Pr\left[Succ_{Adv,GM_{1}}-\frac{1}{2}\right]|.$$With Equations (5) and (6), we derive the below equation:
(8)$$\frac{1}{2}Adv_{t}=\left|Pr\left[Succ_{Adv,GM_{1}}\right]-Pr\left[Succ_{Adv,GM_{4}}\right]\right|.$$Using the trigonometric inequality, we can obtain the results of Equations (4), (5), and (7).
(9)$$\begin{array}{cc} \frac{1}{2}Adv_{t} & =|Pr\left[Succ_{Adv,GM_{1}}\right]-Pr\left[Succ_{Adv,GM_{4}}\right]|\\ \; & \leq |Pr\left[Succ_{Adv,GM_{1}}\right]-Pr\left[Succ_{Adv,GM_{3}}\right]|\\ \; & \;+|Pr\left[Succ_{Adv,GM_{3}}\right]-Pr\left[Succ_{Adv,GM_{4}}\right]|\\ \; & \leq |Pr\left[Succ_{Adv,GM_{1}}\right]-Pr\left[Succ_{Adv,GM_{2}}\right]|\\ \; & \;+|Pr\left[Succ_{Adv,GM_{2}}\right]-Pr\left[Succ_{Adv,GM_{3}}\right]|\\ \; & \;+|Pr\left[Succ_{Adv,GM_{3}}\right]-Pr\left[Succ_{Adv,GM_{4}}\right]|\\ \; & \leq \frac{q_{h}^{2}}{2|Hash|}+\frac{q_{p}^{2}}{2|PUF|}+max\left\{C\cdot q_{send}^{s},\frac{q_{s}}{2^{l_{D}}}\right\} \end{array}$$Finally, multiply both sides of Equation (Equation 8) by 2 to obtain the desired result.
(10)$$Adv_{t}\leq \frac{q_{h}^{2}}{\left|Hash\right|}+\frac{q_{p}^{2}}{\left|PUF\right|}+2max\left\{C\cdot q_{send}^{s},\frac{q_{s}}{2^{l_{D}}}\right\}$$Therefore, we prove Theorem 1. □

### 6.4. Scyther Tool Simulation

In this section, we simulate IoT-PUFTAP using the scyther tool [34]. The scyther tool is a push-button tool to verify and analyze various security protocols. It supports unbounded model checking and multi-protocol analysis and provides a graphical user interface (GUI) to trace security vulnerabilities [49]. We validated the proposed protocol using the scyther tool according to the specifications below:Scyther tool checks security attack classes and possible protocol behaviors of the proposed protocol based on a pattern refinement algorithm.Scyther tool traces the most efficient and optimal security attacks through the “Find best attacks” setting.Scyther tool analyzes the security of the proposed protocol using claim events, including “Secret”, “Alive”, “Weakagree”, “Niagree”, and “Nisynch”.To support multiple executions of protocols in the scyther tool, the “Maximum number of run” and “Maximum number of patterns per claim” parameters are set to 5 and 10, respectively.

To simulate IoT-PUFTAP, we write code in “Security Protocol Description Language (SPDL)”. After that, the scyther tool simulates the “Secret”, “Alive”, “Weakagree”, “Niagree”, and “Nisynch” claim events under the DY model. Note that the claim event “Secret” means that the parameter can ensure confidentiality during the authentication phase. The claim event “Alive” denotes that the participants are alive and running the protocol in same session. “Weakagree” can be satisfied when participants actually communicate with a legal participant. “Niagree” can be guaranteed when participants agree on the exchanged parameters. “Nisynch” is the non-injective synchronization claim event, which means that messages are exchanged from legal participants in appropriate sequence. We conducted simulation on a Ubuntu 20.04.2 LTS virtual machine with an Intel Core i3-8100 3.60 GHz CPU and 16.0 GM of RAM.

#### 6.4.1. Scyther Framework

Figure 4 shows the basic framework of the scyther tool. Firstly, we describe the proposed protocol into the scyther GUI according to the syntax of SPDL. Then, the scyther command-line tool performs the security validation using claim events. Finally, the command-line tool outputs the summary reports and trace class graphs. When the protocol satisfies each claim event, the result window displays the “OK” message and “No attacks” comment.

#### 6.4.2. SPDL Specification

Figure 5 shows the PUFTAP-IoT written in SPDL code. In PUFTAP-IoT, there are three roles: user UI, gateway GWN, and sensing device SDJ. The user UI sends an authentication request message $\left\{Msg_{1},V_{1},T_{1},THID_{i},PSID_{j}\right\}$ to the GWN using the $send_{1}$ function. Then, GWN receives the message using the $recv_{1}$ function and sends $\left\{Msg_{2},V_{2},V_{3},T_{2}\right\}$ to the SDJ. The SDJ computes the session key Skey and returns $\left\{Msg_{3},V_{4},T_{3}\right\}$. The GWN transmits $\left\{Msg_{4},V_{5},V_{6}\right\}$ to the UI.

#### 6.4.3. Simulation Result

Figure 6 indicates the simulation result of PUFTAP-IoT. The result shows that each role is not exposed to security attacks and ensures the “Secret”, “Alive”, “Weakagree”, “Niagree”, and “Nisynch” claim events. Therefore, we can demonstrate that PUFTAP-IoT can resist security vulnerabilities and ensure mutual authentication between each participant.

## 7. Efficiency Analysis

In this section, we compare computation cost, communication cost, and security aspects with other relevant papers to prove the efficiency of PUFTAP-IoT.

### 7.1. Security and Functionality Features Comparison

In this section, we compare PUFTAP-IoT with the related existing protocols in terms of speculation, replay and man-in-the-middle, spoofing, guessing, known session-specific temporary information (KSSTI), device capture, device impersonation, and session key disclosure attacks and security features such as anonymity, forward secrecy, and secure mutual authentication. Table 3 indicates that the existing protocols do not meet all security requirements. On the other hand, PUFTAP-IoT meets all essential security requirements for communication in IoT environment.

### 7.2. Computation Cost Comparison

We cited [50,51] to compare and analyze computation cost with other authentication protocols. Accordingly, we hypothesized notations and times for cryptographic functions and functional functions as follows: $T_{h}$, $T_{rg}$, $T_{puf}$, and $T_{f}$ as the execution time needed for hash function, random nonce generation, PUF function, and fuzzy extraction, where $T_{h}$, $T_{rg}$, $T_{puf}$, and $T_{f}$ are 0.23 ms, 53.9 ms, 12ms, and 2.68 ms, respectively. Table 4 briefly shows the comparison results.

### 7.3. Communication Cost Comparison

In this section, we compare the cost of communication with other authentication protocols. We assume that each value according to [52]: SHA-1 hash digest, entities’ identity, random nonce, and timestamp is 160, 160, 128, and 32 bits, respectively. Based on this assumption, the communication cost of PUFTAP-IoT is analyzed. Messages $\{Msg_{1},V_{1},T_{1},THID_{i},$$PSID_{j}\}$, $\left\{Msg_{2},V_{2},V_{3},T_{2}\right\}$, $\left\{Msg_{3},V_{4},T_{3}\right\}$, and $\left\{Msg_{4},V_{5},V_{6}\right\}$ require (160 + 160 + 32 + 160 + 160 = 592), (160 + 160 + 160 + 32 = 512), (160 + 160 + 32 = 352), and (160 + 160 + 160 = 480) bits, respectively. Thus, the total communication cost requires 592+512+253+480=1837 bits. Table 5 is the analysis of the communication cost consumption of different protocols.

### 7.4. Results of Comparison

The results of the comparative analysis of PUFTAP-IoT and other papers in terms of security, computation cost, and communication cost are as follows. Although PUFTAP-IoT has a higher computational cost compared with authentication protocols in other papers, the communication cost is similar or lower. Moreover, from a security point of view, PUFTAP-IoT is safe against a variety of attacks and can provide security for guessing, brute-force, and device capture attacks using three-factor, PUF, honey_list, etc. Therefore, PUFTAP-IoT can provide very secure services to service users in the IoT environment, even though the computation cost is higher than other authentication protocols.

## 8. Conclusions

With the development of WSN and IoT, areas using IoT are gradually expanding. Therefore, a secure authentication protocol is required to provide secure IoT services. In this paper, we analyze the security vulnerabilities of two-factor and three-factor authentication protocols in various IoT-based environments. To compensate for the security vulnerabilities of these protocols, we proposed PUFTAP-IoT, which applied PUF, honey_list, and three-element technology. We used BAN logic to prove that PUFTAP-IoT can provide secure mutual authentication. We also demonstrated that PUFTAP-IoT can achieve Sean key security through the ROR model. In addition, the scyther simulation tool was used to show that the proposed protocol is safe against various attacks in a wireless network environment. In addition, as a result of the performance analysis of the protocol, it was found that it provides a more secure service in the IoT environment compared with other authentication protocols. In conclusion, PUFTAP-IoT is safer for real-world applications in IoT environments than other related technologies. In the future, based on the proposed protocol, we will analyze the network delay and through put of the protocol through programming and simulation and apply the developed protocol to the real environment to develop better protocols.

## Figures and Tables

**Figure 1 sensors-22-07075-f001:**
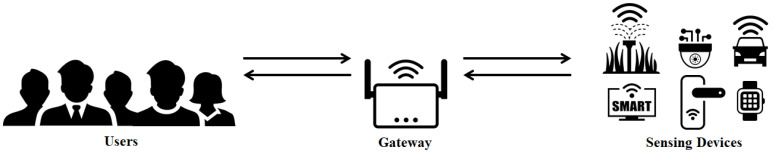
PUFTAP-IoT’s system model.

**Figure 2 sensors-22-07075-f002:**
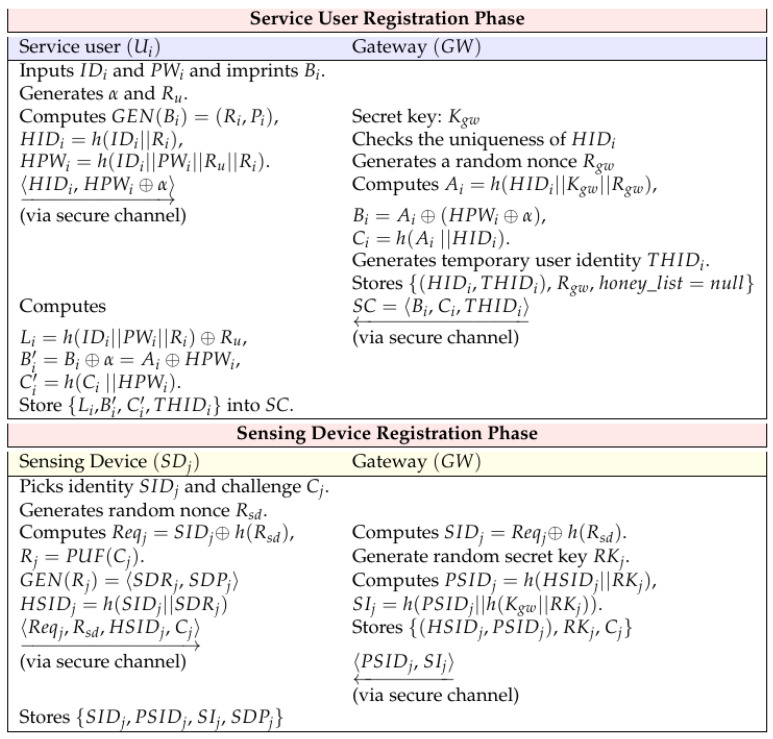
Registration phase.

**Figure 3 sensors-22-07075-f003:**
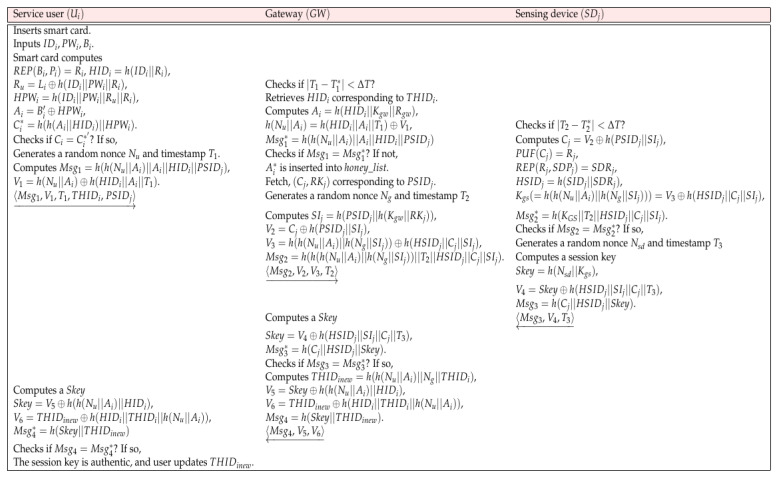
Login and authentication phase.

**Figure 4 sensors-22-07075-f004:**
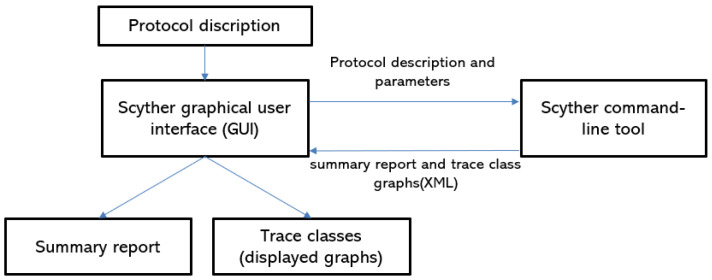
Basic framework of the scyther tool.

**Figure 5 sensors-22-07075-f005:**
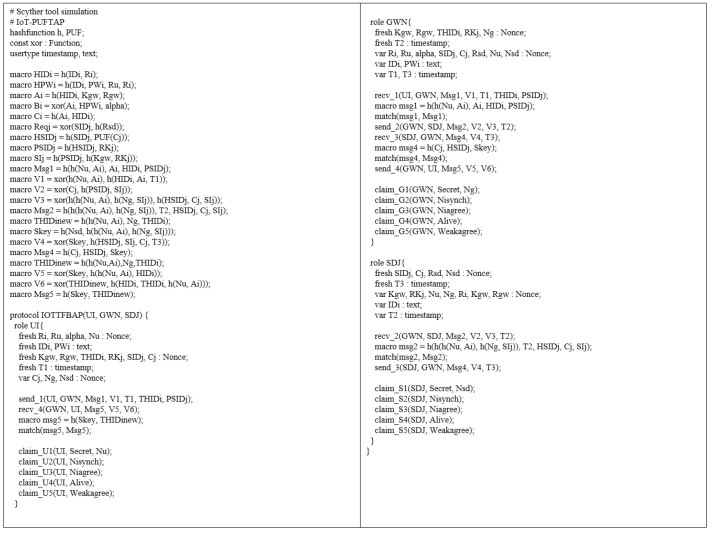
PUFTAP-IoT written in SPDL code.

**Figure 6 sensors-22-07075-f006:**
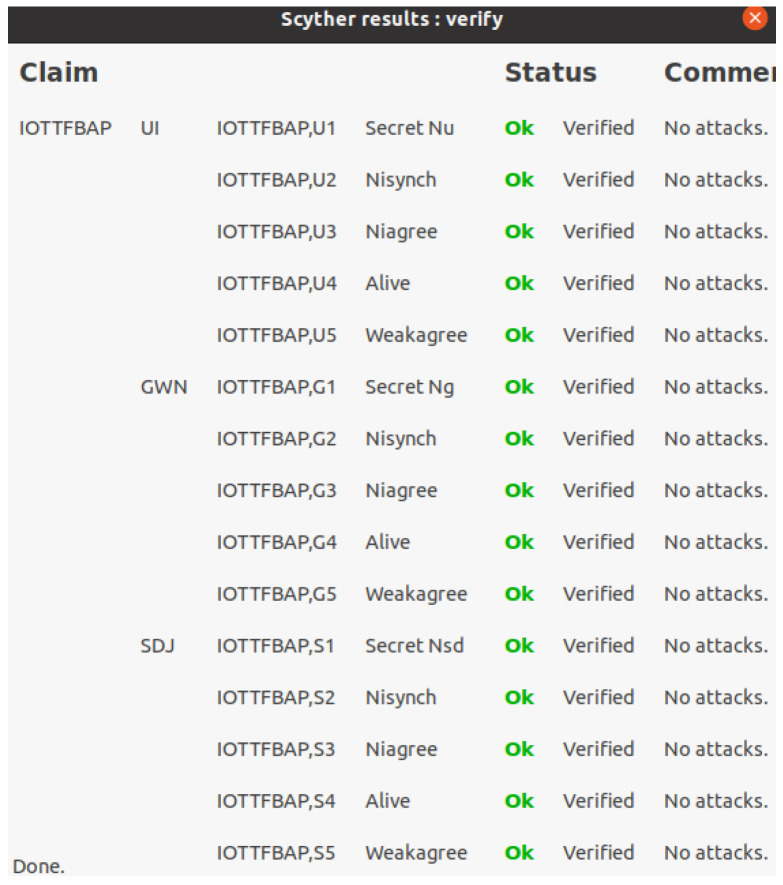
Scyther tool simulation result of PUFTAP-IoT.

**Table 1 sensors-22-07075-t001:** Notation.

Notations	Meanings
$U_{i}$	*i*-th user
$SD_{j}$	*j*-th sensing device
GW	Gateway node
SC	Smartcard
$ID_{i}$	Identity of $U_{i}$
$SID_{j}$	Identity of $SD_{j}$
$PW_{i}$	Password of $U_{i}$
$HPW_{i}$	The hidden password of i-th user
Bi	Biometrics of $U_{i}$
PUF	The Physical Unclonable Function
$C_{j},R_{j}$	The challenge/response pair
GEN,REP	Generation and reproduction algorithm of fuzzy extractor
$K_{gw}$	Secret key of GW
$R_{x}$, $N_{x}$	Random nonces
$T_{x}$	Timestamps
$HID_{i},PSID_{j}$	Pseudo-identity of $U_{i}$ and $SD_{j}$
$THID_{i}$	Temporary user identity $U_{i}$
Skey	Session key
||	Data concatenation operator
⊕	Bitwise exclusive-or operator
h(*)	Collision-resistant one-way hash function

**Table 2 sensors-22-07075-t002:** The basic notations of BAN logic.

Notations	Description
Skey	The session key in PUFTAP-IoT
#ε	The statement *S* is **fresh**
χ|≡ε	χ**believes** the statement ε
χ⊲ε	χ**sees** the statement ε
χ|∼ε	χ once **said** ε
$<\varepsilon {>}_{F}$	ε is **combined** with formula *F*
${\left\{\varepsilon \right\}}_{Key}$	Encrypt ε with Key
χ⇒ε	χ **controls** ε
$\chi \overset{Key}{↔}\omega$	χ and ω shard and use Key for communication

**Table 3 sensors-22-07075-t003:** Security properties comparison.

Security Properties	PUFTAP-IoT	Chunka et al. [16]	Amintoosi et al. [18]	Hajian et al. [27]
Replay attack	o	o	o	o
Man-in-the-middle attack	o	o	o	o
Guessing attack	o	x	x	o
Impersonation attack	o	x	x	x
KSSTI attack	o	x	o	x
Smart card stolen attack	o	x	x	o
Device capture attack	o	o	o	x
Anonymity	o	x	x	o
Perfect forward secrecy	o	o	o	o
Using three factors	o	x	x	o
Using PUF	o	x	x	x
Using honey_list	o	x	x	x
Secure mutual	o	x	x	x
authentication				

x: insecure against an attack; o: secure against an attack.

**Table 4 sensors-22-07075-t004:** Computation cost of login and authentication phase.

Protocol	User	Gateway/Sever	Sensing Device/Sensor	Total Cost
Chunka et al. [16]	6$T_{h}+1T_{rg}$	$5T_{h}+1T_{rg}$	$10T_{h}+1T_{rg}$	$21T_{h}+3T_{rg}\left(166.53\;\mathrm{ms}\right)$
Amintoosi et al. [18]	8$T_{h}+1T_{rg}$	$10T_{h}+1T_{rg}$	$10T_{h}+1T_{rg}$	$23T_{h}+3T_{rg}\left(169.06\;\mathrm{ms}\right)$
Hajian et al. [27]	$12T_{h}+1T_{rg}+1T_{f}$	$10T_{h}$	$5T_{h}+1T_{rg}$	$27T_{h}+2T_{rg}+1T_{f}\left(116.69\;\mathrm{ms}\right)$
Ours	$11T_{h}+1T_{rg}+1T_{f}$	$16T_{h}+1T_{rg}$	$7T_{h}+1T_{rg}+1T_{puf}+1T_{f}$	$34T_{h}+3T_{rg}+1T_{puf}+2T_{f}\left(186.88\;\mathrm{ms}\right)$

**Table 5 sensors-22-07075-t005:** Communication cost of login and authentication phase.

Protocol	Total Communication Costs	No. of Messages
Chunka et al. [16]	2560 bits	4
Amintoosi et al. [18]	1664 bits	4
Hajian et al. [27]	2144 bits	5
Ours	1837 bits	4

## Data Availability

Not applicable.

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
