# Peer review of "PUFTAP-IoT: PUF-Based Three-Factor Authentication Protocol in IoT Environment Focused on Sensing Devices"

_sensors, 2022, doi:10.3390/s22187075_

Round 1
Reviewer 1 Report
In this paper, the authors proposed a PUF-based authentication protocol for IoT environment. In particular, they proposed a protocol focusing on sensing devices and utilized honey_list and three-factor technologies including bio information. The authors performed comparative analysis in terms of security, computation, and communication costs with other batch authentication schemes proposed in WSN and IoT environments.
Overall, the work is novel and it has a solid contribution. The security analysis is sound and also the flow of the work is nice. However, the paper needs minor revisions prior to publication. Some comments are given below:
1. Various symbols exist in this paper. Therefore, sufficient explanation of the symbols is necessary for readers to clearly understand this paper. For readers' readability, I suggest that the authors add some more symbols shown in Table 1.
2. I suggest that the authors discuss future work in Section 8 Conclusions.
3. The authors should carefully proofread the paper for some typos and grammatical errors.
Reviewer 2 Report
The paper proposes a authentication protocol for sensing devices in IoT environments. In general, the theoretical and proposal part of the work seems to be sufficiently mature. The organization of the paper is well structured. Before acceptance, I suggest the following improvements:
1) I think that a good section of related work is mandatory in a journal paper. The section of related work shows knowledge of authors in the related area and emphasizes on the significance and originality of the paper. Regarding safety and reliability of the IoT system (i.e., cascading failures), some important work has emerged in the past two years. The author should further enrich this part, at least mention them in the introduction part, such as “Analysis on Cascading Reliability of Edge-Assisted Internet of Things”, “Cascade Failures Analysis of Internet of Things under Global/Local Routing Mode”.
2) The information and details about the simulation framework are not in the paper or they are too disseminated. A section about the experimental framework is mandatory. For example: number of simulations, representativeness of these scenarios, heterogeneity of nodes and links,...
3) The paper would be much stronger focusing the discussion about the interpretation of that data and abstracting out and elaborating lessons and knowledge that could apply beyond the specific case.
4) Some typos occur, that can be easily fixed with a proofreading.
5) I suggest the authors to reorganize the first section-Introduction. In this part, the authors list several subsections. I think it is not necessary. If possible, the authors should combine them into one part. Regarding how to write this part, the authors can reference "Analysis on Invulnerability of Wireless Sensor Networks based on Cellular Automata".
Round 2
Reviewer 2 Report
Can be accepted.